# Vulnerability Analysis of the Riumar Dune Field in El Garxal Coastal Wetland (Ebro Delta, Spain)

Inmaculada Rodríguez-Santalla *, Alejandro Díez-Martínez and Nuria Navarro

Departamento de Biología, Geología, Física y Química Inorgánica, ESCET—Universidad Rey Juan Carlos, C/Tulipán s/n, 28933 Madrid, Spain; alexhogun@gmail.com (A.D.-M.); nuria.navarro@urjc.es (N.N.)
* Correspondence: inmaculada.rodriguez@urjc.es

**Abstract:** The aim of this work is to apply a vulnerability index in the dune field located in the Riumar urban zone at the mouth of the Ebro River. This dune field represents the natural barrier of the El Garxal coastal lagoon system. The index used integrates the dimensions of exposure, susceptibility, and resilience from the analysis of 19 variables. The results obtained show moderate susceptibility and high resilience, which are in line with the behavior of this dune field during the last sea storms (Gloria in January 2020 and Philomena in January 2021, among others) that have tested the capacity of this system to cope with the effects of these storms. Therefore, increasing the knowledge of the factors affecting the vulnerability of the dunes can be helpful in the management and conservation of these coastal environments.

**Keywords:** dunar vulnerability index; dunar susceptibility; dunar resilience; coastal lagoon system

## 1. Introduction

The systems formed by the coastal barriers and lagoons correspond to relatively shallow areas that have been partially or completely isolated from the sea due to the development of spits or sand barriers caused by the wave and tide dynamics. Kjerfve [1] defines the coastal lagoon as an inland water body, usually oriented parallel to the coast, separated from the ocean by a barrier, connected to the ocean by one or more restricted inlets, and having depths that seldom exceed a couple of meters. Otvos [2] established a coastal barriers classification according to the predominant processes and specifically defined the term barrier as emergent coastal-nearshore landform group represented by shore-parallel elongated islands, often in chains, barrier spits, and mainland strandplains, including chenier ridge clusters.

Furthermore, the coastal lagoons that remain protected represent areas of high ecological and environmental value. In these lagoons, there is a mixture between continental and marine water, representing environments with a great diversity of habitats and communities, with an exceptional capacity as $CO_2$ sinks [3], and they are considered sentinel ecosystems for global change by FAO [4].

The development and maintenance of the lagoon are associated with the persistence of the barrier that protects them. Precisely, among the main functions that Otvos [2] attributes to the barriers are that of harboring and protecting the ecological habitats against storm destruction.

Reinson [5] describes different sub-environments developed on a barrier island, including the backshore dune deposit. The development of dune fields provides consistency, and consequently provides greater protection to the back wetland. According to Rodríguez-Santalla [6], the dune systems are of great importance to the coastal areas because they represent the best defense against strong waves produced by storms. Besides, from an ecological approach, the dunes are home to numerous plant species, and the habitat of several animal species. Its degradation and disappearance represent an enormous biodiversity

and ecosystem loss, whose functions and services can only be partially recovered and at a high economic cost through restoration programs [7].

Given the important role of the coastal dunes in maintaining the coastline, several indexes have been developed [8–11], which analyze the vulnerability of dune fields in order to establish the factors that endanger their conservation. The vulnerability analyses of dunes integrate the interaction between aeolian and marine processes, geomorphology, vegetation, and human pressure [10]. García-Mora et al. [11] established an index (Dunar vulnerability index; DVI) that brings together a set of variables grouped into five groups: geomorphological conditions of the dune system (GCD), marine influence (MI), aeolian effect (AE), vegetation condition (VC), and human effect (HE). Finally, the DVI is computed as the unweighted average of the five partial vulnerability indexes.

Although the DVI has been frequently used [7,12–16], an agreement on how many variables must be pooled into any vulnerability index and whether each variable should be weighted or not has still to be achieved [15]. García-Mora et al. [11] consider that a good index should be based on the minimum amount of necessary information. Accordingly, the index has been adapted to the environmental conditions where the dune fields are developed, such as the Atlantic coastal environments [11,17], the Mediterranean dune vulnerability index (MDVI) for sandy coasts [15], and the vulnerability index of arid beach–dune systems [16]. In this latter case, the variations introducing by Peña et al. [16] are related not only to the definition of variables but also to the way to express vulnerability from three analytical dimensions: exposure, susceptibility, and resilience. According to Smit and Wandel [18], vulnerability is a function of the exposure and sensitivity to hazardous conditions, and the ability, capacity, or resilience of the system to cope with, adapt, or recover from the effects of those conditions. Therefore, the function of the vulnerability index is to simplify several complex and interacting parameters, represented by diverse data types, to a more readily understood form and, therefore, has greater utility as a management tool [19].

The purpose of this study is to apply a methodology based on the index of García-Mora et al. [11] and Peña et al. [16] that allows analyzing vulnerability in terms of exposure, susceptibility, and resilience of the dune field located in the mouth of the Ebro River, which protects the El Garxal wetland. This arrangement will determine the most sensitive variables towards its protection against increasingly frequent storm damage (e.g., storm Gloria in January 2020 or storm Philomena in January 2021) and allow the preservation of the El Garxal coastal lagoon system.

## 2. Materials and Methods

### 2.1. Study Area

The El Garxal wetland is located in the final stretch of the mouth of the Ebro River, in the zone of the Riumar beach (Figure 1). This river has developed a delta, which constitutes one of the main areas in the Mediterranean basin. The left bank of the mouth of the river is one of the Ebro delta areas where accretion processes are most evident. In contrast, the Tortosa Cape is the most vulnerable zone of the delta to coastal erosion due to its exposition to the highest energetic and persistent waves coming from the East and NE. In fact, this area has recorded the most drastic geomorphologic changes in recent decades. Winds blowing from the NW are much stronger and generate a wind-driven littoral drift current flowing to the South, although the limited fetch generates not very energetic waves [20]. The Ebro Delta has a microtidal regime, characterized by maximum astronomical and meteorological tides of 0.25 m and 1 m, respectively [21]. The coastline evolution has been studied by numerous researchers ([20–25], among others).

The Ebro river delta has gone through quite different stages of development [24]. However, the biggest and most abrupt changes occur contemporaneously with the construction of several dams during the 20th Century, particularly those of Mequinenza, Ribarroja, and Flix, in the lower reach of the Ebro River. Due to an intense river flooding episode in 1937, a

new mouth was opened in 1937, which remains to this day [21]. The former mouth became almost inactive and by 1946 had been completely closed by sand deposition.

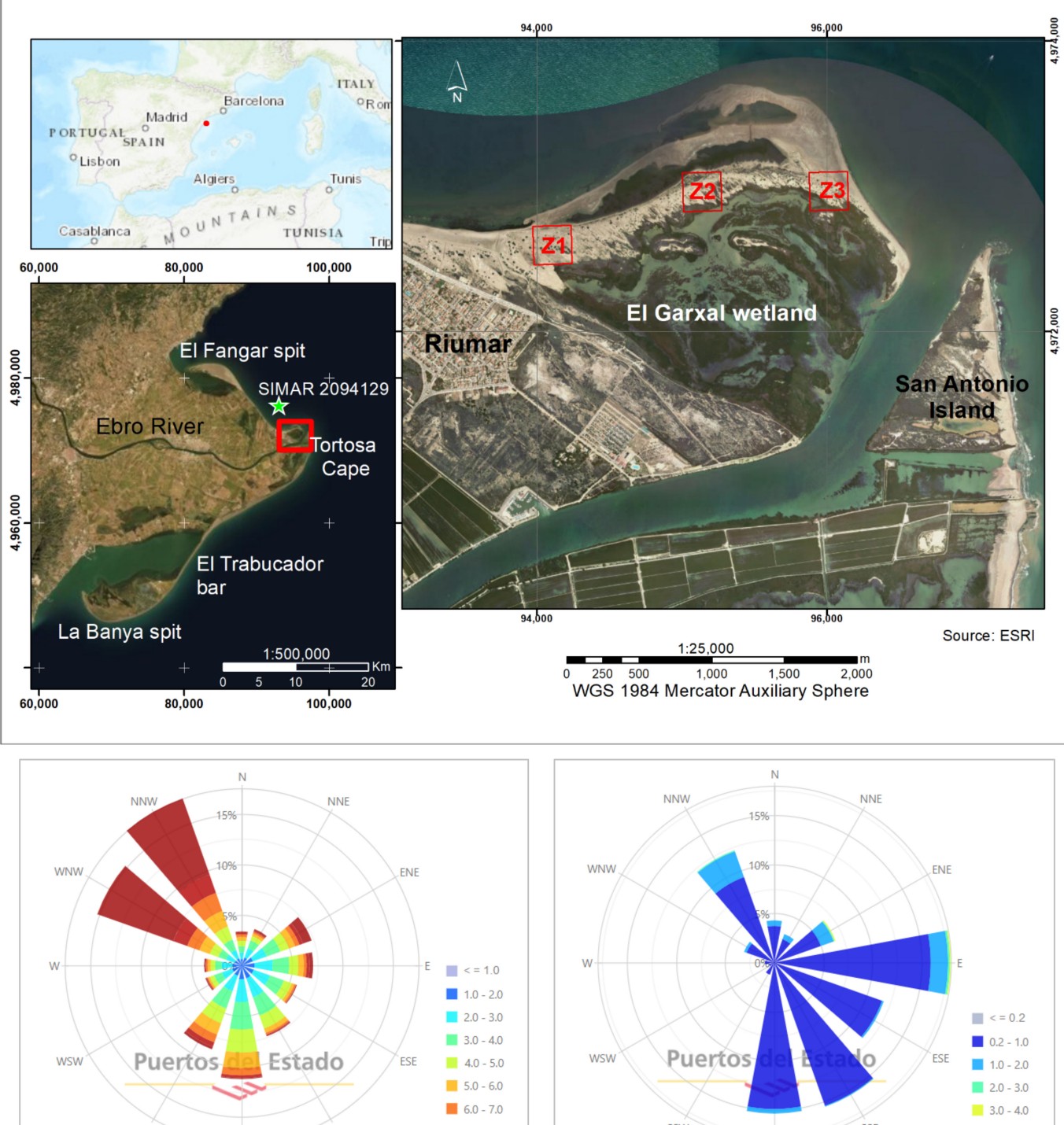

**Figure 1.** Map showing the location of the El Garxal coastal wetland and the Riumar dune field. The red squares show the three sites selected (Z1, Z2, and Z3). The bottom left chart shows the Wind rose (average wind speed (m/s)), and the right chart, the Wave rose (significant wave height (m)) of SIMAR Point 2094129 (the green star in the left-hand graph). Period: 2010–2018. Efficiency: 99.83%. Reproduced from Puertos del Estado (http://www.puertos.es/ (accessed on 2 March 2021)).

The contribution of sediments by the river, and the input of sediment arrived by the main littoral drift transport, together with the protection against the waves that Cape Tortosa offers, has favored the deposit of beach ridges that have stabilized while being colonized by vegetation. Thus, this sector has prograded, giving rise to a more complex barrier-lagoon system formed by the beach on the external coast, where the Riumar dune system is installed, and the El Garxal coastal lagoon in the interior. An analysis of the geomorphological changes identified in the Ebro River mouth was carried by Ramírez-Cuesta et al. [20] (Figure 2). According to Otvos [2], the study zone corresponds to a Barrier spit on which the dune field is developed.

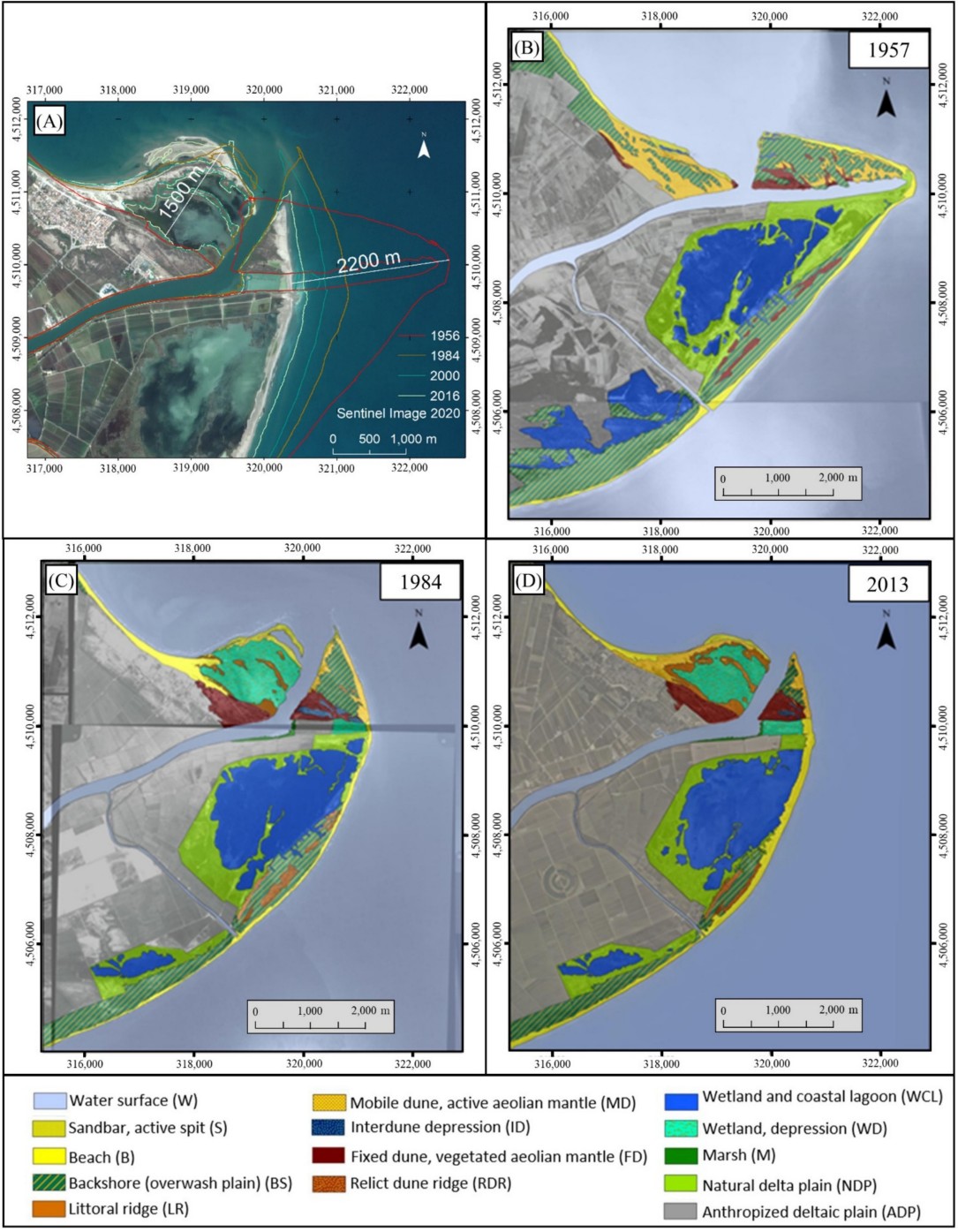

**Figure 2.** (**A**) Coastal evolution of Ebro River mouth (coastal lines on Sentinel image T31TCF_20201101T105209_TCI). (**B–D**) Geomorphological features evolution, reproduced from Ramírez-Cuesta et al. [20].

The formation and evolution of these dune fields have been described by Sánchez-García et al. [26], and Rodríguez-Santalla and Treviño [27]. The origin of the Riumar dune field is related to changes in the river mouth. The sand of the dune comes from the mobile dunes of El Fangar spit (in the north of Ebro delta), which is transported by the main wind direction NW–SE. The orientation of the dune is consistent with the main wind direction. Rodríguez-Santalla and Treviño [27] carried out a comparison of the amount of sediment reached by the dune field between 2011 and 2016. The results showed an increase in the dune field surface area and volume, as well as the maximum dune height reached. Furthermore, an increase in deposits has been evidenced inside the El Garxal coastal lagoon caused by the transfer of sediments from the dune towards the lagoon.

### 2.2. Data Set

The variables used have been derived from different sources: Digital elevation model (DEM) from LiDAR data belonging to the Cartographic and Geologic Institute of Catalonia (ICGC), with a density of 0.5 points/m$^2$; and orthophotos from the National Geographic Institute (IGN). The software used has been ArcGIS 10.x. The GIS constitutes a highly valuable tool oriented to land management and planning [20,28]. In addition, sediment samples taken from the dune have been analyzed to get the textural characteristics of sediments.

The dune field has been extracted from LiDAR data. The base height of the dune was established at 1 m above sea level. According to the degree of exposure of the dune, three sites of 200 m × 200 m distributed along the dune system have been selected (Figure 1).

The tool Iso Cluster Unsupervised Classification of ArcGIS has been used to separate the vegetation cover from the orthophoto of 2018 [29]. This method reclassifies the image according to the number of classes defined by the user. In this study, two classes were defined, on the one hand, the Riumar urban zone and bare sand, and on the other, the vegetated surface.

In addition, the extension to ESRI ArcGIS DSAS has been used to analyze the coastal evolution, which generates perpendicular transects to the reference baseline, measures the distance between the baseline and each shoreline intersection along a transect, and combines date information and positional uncertainty for each shoreline to calculate rate-of-change statistics from multiple historical shoreline positions [30].

### 2.3. Definition of Variables

According to García Mora et al. [11], Ley et al. [7], and Peña-Alonso et al. [16], five groups of variables have been identified, and another group was integrated (*): marine influence (MI); human pressure (HP); characteristics of the vegetation cover (VC) geomorphology of the dune system (GD) and geomorphology of the beach (GB*); and aeolian influence (AI). The choice of these variables has been conditioned by physical and geomorphology characteristics of the dune system, assuming as a fundamental principle to use the minimum amount of necessary information [11]. The variables have been organized following the structure of Peña-Alonso et al. [16] to establish the exposure (EXP: MI and HP), susceptibility (SUS: VC and GD), and resilience (RS: GB and AI) indicators. To achieve this, it was necessary to divide the geomorphological variables into two categories: geomorphology of the dune system and geomorphology of the beach. All variables have been normalized between 0 to 4 values, where 0 implies the minimum and 4 the maximum value (Table 1).

#### 2.3.1. Marine Influence (MI)

The significant wave height (Hs) and the tidal range, both obtained from Grases et al. [31] define the characterization of the marine influence. These factors determine the sediment availability of the dune [7] and its seaward development, which is limited by storm tide height that may cause undercutting of the dune face or washovers [11].

### 2.3.2. Human Pressure (HP)

This factor takes into account the activities of the visitors and the managers. According to García Mora et al. [11] human impacts on coastal dunes can be temporary (e.g., pedestrian, vehicle, and animal trampling) or permanent roads, housing, parking, crops and, forestry) depending on the activity or use developed in the dune field. The parameters considered have been the pressure and frequency of visitors (from Ortells and Querol, [32]) and access difficulty, which is obtained from field observations of the presence of pedestrian walkways over the dunes, information panels, and access controls.

### 2.3.3. Characteristics of the Vegetation Cover (VC)

The vegetation of coastal dunes plays an important role in stabilizing the surface against wind erosion and provides a habitat for wildlife [7]. The coastal dune plants belong to three specific functionally based types [10]:

Type I consists mainly of winter annuals, small size and are soft-leaved, with no presumed adaptations to the dune environment, prone to wave erosion.

Type II is mostly perennials with a below-ground root network and leaves with adaptations to coastal environmental stress.

Type III includes plants capable of being dispersed by seawater, which are able to withstand burial. Types I and II can stand disturbance only.

The vegetation parameters considered have been the average vegetation cover and the area with vegetation type II (the prevailing in the area). The calculation of the average vegetation cover has been done on the three sites selected, considering formations greater than 5 m$^2$, by the intersection between the vegetation cover and the shape with the sites. The percentage of the area with vegetation was estimated from the dune system and the vegetation cover for the three sites.

### 2.3.4. Geomorphology of the Dune System (GD)

The length and the width of the active dune system (both in km) were obtained from LiDAR data.

Modal height (the average height of the dune system from the base to the top) was previously established at 1 m AMSL. The dune height was obtained from LiDAR data using some ArcGIS tools.

The particle size of the windward slope of the dune was obtained from the sieved in the laboratory of sand samples collected during some field works in 2017 and 2020.

Relative surface with scarps or erosion: scarping is common on coastal dunes and can be defined as basal erosion and undercutting of the stoss slope of a dune due to wave attack or possibly stream and river erosion [33]. Although during fieldwork and in the preliminary visual analysis of the orthophotos, no escarpment was observed in the entire study area, a raster layer has been created with slope values greater than 50°. Additionally, attention has been paid to the effects of the major storms that have occurred in recent months (Gloria, in January 2020 and Filomena in January 2021).

The modal state of the beach is the most frequent morphology that a beach presents, and there are two extreme states [7]: dissipative and reflective. The dissipative beach presents high wave energy, a wide surf zone with one or more shore-parallel bars, and a wide, low gradient intertidal beach composed of fine sand. The reflective beach shows low wave energy, and a beach face with high slopes formed by coarser sediment. There is no bar in surf zone. Intermediate states exist associates to open coasts, with moderate waves and fine to medium sand, and present some bars.

### 2.3.5. Geomorphology of the Beach (GB)

The number of submerged or emerged beach ridges has been obtained from Rodríguez-Martín and Rodríguez-Santalla [34].

The dry beach width has been calculated using the ArcGIS extension DSAS V5.0 [30]. The particle size of the dry beach sediment has been obtained from Grases et al. [31].

In addition, the shoreline variation has been derived from the Net Shoreline Movement (NSM) for the years 1998 and 2018 using DSAS extension for ArcGIS. The same tool was used to obtain the dry beach surface variation.

### 2.3.6. Aeolian Influence (AI)

To obtain the sand supply input, an analysis of the volume variation was carried out by comparing 2011 and 2016 MDTs using the cut and fill tool of ArcGIS, following the methodology of Rodríguez-Santalla and Triviño-Monzón [27] and Rodríguez-Santalla et al. [35].

**Table 1.** Variables considered that were reproduced from García Mora et al. [11], Ley et al. [7], and Peña-Alonso et al. [16]. All variables were normalized between 0 to 4 values, where 0 implies the minimum and 4 the maximum value. Legend: MI: marine influence; HP: human pressure; VC: vegetation cover; GD: geomorphology of the dune system; GB: geomorphology of the beach: AI: aeolian influence; EXP: exposure; SUS: Susceptibility; RS: Resilience.

| | | Variables | 0 | 1 | 2 | 3 | 4 | Source |
|---|---|---|---|---|---|---|---|---|
| **EXP** | **MI** | Waves Intensity (Hs) | <0.55 | 0.55–0.85 | 0.86–1.05 | 1.06–1.25 | >1.25 | [31] |
| | | Tidal range (m) | <2 | - | 2 to 4 | - | >4 | |
| | **HP** | Visitor pressure | Low | - | Moderate | - | High | Field/[32] |
| | | Visitor frecuency | Low | - | Moderate | - | High | |
| | | Access difficulty | High | - | Moderate | - | Low | |
| **SUS** | **VC** | Average vegetation cover (m$^2$) | >230 | <230 | <125 | <60 | <10 | Field/GIS |
| | | Percentage of the area with vegetation (Type II) | <5 | <15 | <30 | <60 | >60 | |
| | **GD** | Active dune system length (km) | >20 | >10 | >5 | >1 | >0.1 | GIS |
| | | Active dune system width (km) | >2 | >1 | >0.5 | >0.1 | <0.1 | GIS |
| | | Average height of the coastal dune (m) | >2 | 1.5 to 2 | 1 to 1.5 | 0.5 to 1 | <0.5 | GIS |
| | | Particle size of the windward slope of the dune ($\Phi$) | $\leq -1$ | 0 | 1 | 2 | 3 | Field/Lab |
| | | Relative surface with scarps (m$^2$) | 0 | <5 | <20 | <50 | >50 | GIS |
| | | Modal beach state | Reflective | - | Intermediate | - | Dissipative | [31] |
| **RS** | **GB** | Number of sandy or rocky bars submerged or emerged | 0 | - | 1 | - | >1 | GIS |
| | | Beach width (m) | 0 | <10 | <25 | <75 | >75 | GIS |
| | | Particle size of the dry beach | <0 | - | 0 a 2 | - | >2 | [31] |
| | | Net Shoreline Movement (m) | $<-40$ | $-10$ to $-40$ | - | $>0$ to $-10$ | $\geq 0$ | GIS |
| | | Beach surface variation | $<-0.4$ | $-0.06$ to $-0.4$ | - | $<0$ to $-0.06$ | $\geq 0$ | GIS |
| | **AI** | Sediment supply input | Low | - | Moderate | - | High | GIS |

### 2.4. Dune Vulnerability Index (DVI)

The dune vulnerability index has been obtained from the unweighted average of the partial vulnerability indices (Is) of each group of variables, which are calculated as the ratio between the sum of the assigned values (Vi) and the sum of the maximum possible values of each group of variables (Vp max):

$$Is = Vi/Vp \; max \qquad (1)$$

Each index is ranged between 0 and 1. Finally, according to Peña-Alonso et al. [16], DVI is:

$$DVI = (SUS*EXP)/RS \qquad (2)$$

## 3. Results

Table 2 compiles the values achieved by each of the variables and for each of the three areas considered (Z1, Z2, and Z3, Figure 1).

**Table 2.** Values assigned to each variable according to the results obtained in the analysis (Section 3.1) for each of the three areas considered (Z1, Z2 and Z3). Legend: MI: marine influence; HP: human pressure; VC: vegetation cover; GD: geomorphology of the dune system; GB: geomorphology of the beach; AI: aeolian influence; EXP: exposure; SUS: Susceptibility; RS: resilience.

| | | Variables | Z1 | Z2 | Z3 | Average |
|---|---|---|---|---|---|---|
| **EXP** | **MI** | Waves Intensity (Hs) | | 4 | | 4.00 |
| | | Tidal range (m) | | 2 | | 2.00 |
| | **HP** | Visitor pressure | | 0 | | 0.00 |
| | | Visitor frecuency | | 0 | | 0.00 |
| | | Access difficulty | | 0 | | 0.00 |
| **SUS** | **VC** | Average vegetation cover (m$^2$) | 3 | 1 | 1 | 1.67 |
| | | Percentage of the area with vegetation (Type II) | 1 | 3 | 2 | 2.00 |
| | **GD** | Active dune system length (km) | | 3 | | 3.00 |
| | | Active dune system width (km) | | 3 | | 3.00 |
| | | Average height of the coastal dune (m) | 1 | 3 | 1 | 1.67 |
| | | Particle size of the windward slope of the dune ($\Phi$) | | 3 | | 3.00 |
| | | Relative surface with scarps (m$^2$) | | 4 | | 4.00 |
| | | Modal beach state | | 4 | | 4.00 |
| **RS** | **GB** | Number of sandy or rocky bars submerged or emerged | | 4 | | 4.00 |
| | | Beach width (m) | | 4 | | 4.00 |
| | | Particle size of the dry beach | | 2 | | 2.00 |
| | | Net Shoreline Movement (m) | | 4 | | 4.00 |
| | | Beach surface variation | | 4 | | 4.00 |
| | **AI** | Sediment input from the primary dune | | 2 | | 2.00 |

### 3.1. Analysis of the Variables

#### 3.1.1. Marine Influence (MI)

According to Grases et al. [31], the Riumar area is a dissipative beach featuring a mild slope at the front part of the beach (0.07%). The Ebro Delta is a wave-dominated micro-tidal environment with a tidal range of approximately 0.25 m [31]. The influence of the astronomical tide and the meteorological tide (atmospheric pressure and sea elevations caused by the wind and the waves) can produce variations of about one meter [36]. The significant wave height (Hs) is 0.75 m, the mean wave period (Tm) is 3.9 s [23], and the storm waves can be exceeding 2 m [37]. The eastern wave component, the higher and more energetic waves, is the predominant cause of morphological changes [6].

#### 3.1.2. Human Pressure (HP)

According to the Statistical Institute of Catalonia on 1 January 2020, the population of the Riumar urban area is 270 habitants (https://www.idescat.cat/poblacio/?q=Riumar&lang=es (accessed on 11 April 2021)). Although the little urban area of Riumar is a touristic zone, the number of visitors is low. According to Aranda et al. [38], this area did not suffer remarkable changes in anthropic features due to its qualification as a Natural Park in 1989.

### 3.1.3. Characteristics of the Vegetation Cover (VC)

The results obtained differ slightly for each zone (Table 3). The most abundant species is *Crucianella maritima*, which is classified as Type II according to García-Mora et al. [39]. Plants of Type II are mostly perennials with a below-ground spreading root network and leaves with presumed adaptations to coastal environmental stress [39].

**Table 3.** Results obtained for the vegetation cover (VC) variables, and for each studied area (Z1, Z2, and Z3).

| Variables VC | Z1 | Z2 | Z3 |
|---|---|---|---|
| Average vegetation cover ($m^2$) | 54 | 192.77 | 179.22 |
| Percentage of the area with vegetation (Type II) | 11 | 39.31 | 27.36 |

### 3.1.4. Geomorphology of the Dune System (GD)

The length and width of the active dune system reflect a high susceptibility value in all dunar systems. The average height of the coastal dune varies in each area. Z1 presents an average height of 1.56 m, with high elevation in its central region, reaching a maximum of 7.3 m ASL; Z2 shows an average height of 0.67 m and a maximum of 4.31 m; and Z3 has higher dunes that reach 7.75 m, with an average height of 1.45 m.

All samples sieved in the laboratory have a medium grain size.

The analysis of the calculated slopes has confirmed that there is no erosive scarp on the entire surface studied. However, once the effects of the last storm events on the eastern part of the dune are known, a high susceptibility value has been granted.

### 3.1.5. Geomorphology of the Beach (GB)

Figure 1 shows sandy hooks developed in front of the dune field. These hooks slowly connected to the land and enabled coastal development [21,38]. Moreover, these sand bars favoring the presence of a wide beach, which has been quantified in 116 m on average. According to Grases et al. [31], the particle size of the dry beach sediment is 200 μm. The net shoreline movement between 1998 and 2018 showed a mean value of +160 m, equivalent to 8 m of advance per year. The surface variation between 1998 (113 $m^2$) and 2018 (300 $m^2$) was 2.65.

### 3.1.6. Aeolian Influence (AI)

The results obtained for dune volume variation between 2011 and 2016 showed a slightly positive trend, going from 546,577 $m^3$ in 2011 to 574,483 $m^3$ in 2016, which represented an increase of 5.11% of the total volume.

### 3.2. Dune Vulnerability Index Assessment (DVI)

Figure 3 shows the results of the partial vulnerability indices (Is) of each group of variables and the DVI final. In addition, a radar chart was obtained illustrating these results. The DVI ranges between 0 and 1, and in accordance with García-Mora et al. [11], as the index increases, the ability of a dune system to withstand further pressure decreases.

| | | Is | | ADVI |
|---|---|---|---|---|
| EXP | MI | 0.75 | 0.3 | 0.25 |
| EXP | HP | 0 | 0.3 | 0.25 |
| SUS | VC | 0.46 | 0.7 | 0.25 |
| SUS | GD | 0.77 | 0.7 | 0.25 |
| RS | GB | 0.9 | 0.83 | 0.25 |
| RS | AI | 0.5 | 0.83 | 0.25 |

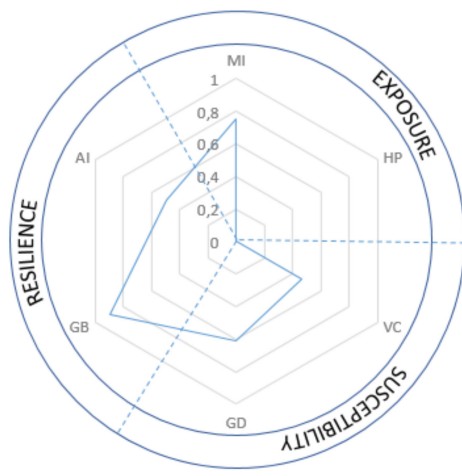

**Figure 3.** (**Left**) Table with partial vulnerability indices (Is) and the DVI calculated. (**Right**) Radar graphic. Legend: MI: marine influence; HP: human pressure; VC: vegetation cover; GD: geomorphology of the dune system; GB: geomorphology of the beach; AI: aeolian influence; EXP: exposure; SUS: susceptibility; RS: resilience.

## 4. Discussion

### 4.1. Exposure Assessment

The dunar system examined has characteristics that, combined with the environment in which they develop, give a very low vulnerability. Their position with relation to the main waves, and the coast orientation, make the main transforming forces of the coast lose some of their capacity. Besides, the storms have a seasonal pattern. According to Jimenez et al. (1997) [23] the energetic period covered the period from October until March, where the eastern waves are dominant; from March to June there is a transition period, and the mild period from June to September is characterized by the lowest wave heights and shorter periods, being south the most frequent direction. During the transition and mild wave action, the shoreline accretion occurs [23] and, therefore, recovers from winter storms. According to Mendoza and Jimenez [40], the most frequent storm category in this area, based on wave energy content, is weak, with a mean duration short (12 h) and mean Hs of 2 m. Occasionally, severe (Hs = 4 m and Tp = 9 sg) and extreme (Hs = 6 m and Tp = 11 sg) storms have happened. One of the most extraordinary was storm Gloria (Figure 4) that occurred on 19 and 23 January of 2020, where a Hs of 7.6 m and H max of 12 m was reached; and the storm surge varied between 50 and 70 cm [41]. From 6 to 11 January 2021, another stormy episode took place (storm Filomena) with wind speeds of up to 70 km per hour and a storm surge of 4.6 m high and period of 10 sg. Figure 4 shows two situations of these events and it can be observed as the dunar body remained despite the very adverse conditions, although the part most exposed to waves suffered an overwashing.

The other variable considered in the exposition term is human activity, which does not seem to exert high pressure over the dunar complex. The population of the Riumar urban zone is relatively low and is very concerned whit erosive and environmental problems of the Ebro Delta. Moreover, the most common visits to the Ebro Delta take place in a single day [32], and it is included within the so-called ecotourism. In the last few years, the Ebro Delta has been configured as a sustainable tourism destination, recognized in the European charter of Sustainable Tourism (CETS) [42]. In addition, the access difficulty to the dune is great and is limited between April to June, coinciding with the nesting period.

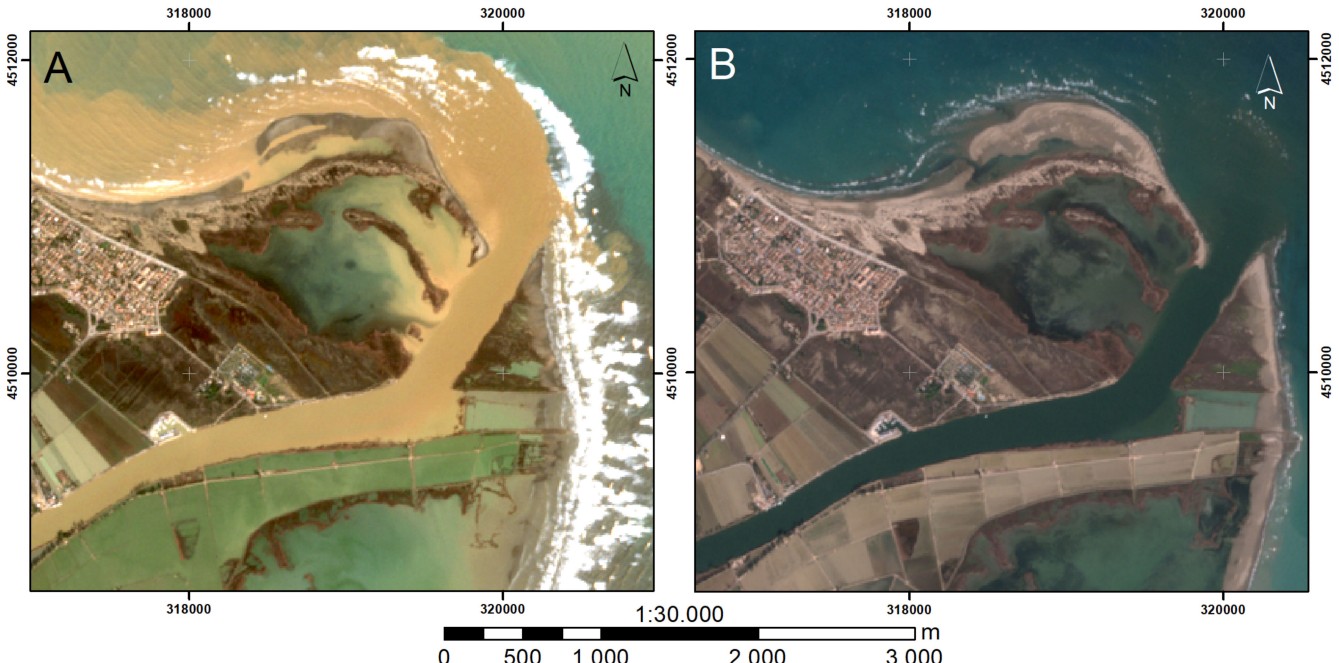

**Figure 4.** Images of Ebro River mouth. (**A**) Sentinel image of 23 January 2020 during storm Gloria. (**B**) Sentinel image of 17 January 2021 after storm Filomena. Reproduced from https://scihub.copernicus.eu/dhus/#/home (accessed on 27 January 2021)).

In terms of hazard exposure, even if one considers the most severe values of waves and tides, and consider the effects of human activity low or null, the significance is low.

### 4.2. Susceptibility Assessment

The evolution of the Riumar field dune can be observed in Figure 2. According to Ramírez-Cuesta et al. [20] and Sánchez-García et al. [26], the dunar system of Riumar acts as a sediment sink, which is transported by the northwesterly winds from the El Fangar spit and is spreading progressively eastwards as the beach progress.

The geomorphology characteristics of the dune (length, width, height, and size of sediments) reveal a stable dune, especially in zone 1 (Z1). In addition, the vegetation Type II is not susceptible to being dispersed by seawater and is characteristic of semi-stabilized dunes, and they favor the fixation of the sands, reducing the effects of wind deflation [7].

According to Mendoza and Jiménez [40], a general rule for a given storm, a dissipative beach is potentially more vulnerable to inundation. Nevertheless, the beach ridges located in front of the dune (Figure 4) have held back part of the wave energy. The most susceptible area that shows dune scarp due to wave storm is the eastern part, which is faced with the waves.

The susceptibility registers relatively high values, although it must be recognized that the situations more extreme were considered.

### 4.3. Resilience Assessment

The beach ridges play a fundamental role in the generation of dune systems [35], reducing the wave energy and increasing the beach width, giving the space to absorb sea-level variations. These deposits constitute a natural source of sediments to build the winter profile, diminishing the energy of the surge [6]. As seen from Figure 2, the beach has an accretionary evolution reaching 1600 m in 60 years. According to Rodríguez-Santalla and Triviño-Monzón [27], the sediment deposited on the mouth area has several sources. Firstly, from the East by erosion of the Buda Island, that gives rise to the beach accretion by beach ridges; and secondly from the northwest, by aeolian transport and longshore currents of the sediment belonging to the dunes of the El Fangar spit and the beaches located in the

north of the mouth. In line with Barrio et al. [43], there exists a relationship between the sedimentary exchange of the beach–dune systems of El Fangar spit and the evolution of the coastline in the Riumar area. The greater the amount of sediment accumulated in the coastal system, the greater the capacity of absorption of the impact, and the more stable the dune complex will be [7].

Guillén and Palanques [44] detected a coarsening trend in the sediments of the Ebro delta coast at a medium-term scale (two decades) that, according to Jimenez et al. [45], should decrease the intensity of erosion hazard.

In respect of aeolian influence, an increase in the height and volume of dune ridges has been observed, which is in line with the results of Ramirez-Cuesta et al. [20] y Rodriguez-Santalla y Triviño-Monzón [27]. As mentioned above, the sediment comes from the El Fangar spit, transported by the northwest winds, which is reflected in the orientation of the dunes (Figure 4).

The conditions of the beach and the state of the dune encourage a high value of resilience.

### *4.4. DVI Assessment*

Taking all variables examined into account, the resilience values are very high and compensate those obtained both in exposure and susceptibility, giving a very low DVI value. Actually, the environmental context in which this dune body develops should be broadened in the analysis. As has been stated, the El Garxal coastal lagoon system is protected by Cape Tortosa, which is the area that currently has the largest rate of erosion of the whole of Ebro Delta. According to Rodríguez-Santalla and Triviño-Monzón [27], as long as Cape Tortosa remains and the system continues to receive sediment, the system can remain. On the other hand, it appears that there is an increase in the frequency and/or intensity of storm events as well as a potential sea-level rise which will affect the present morphology of the delta [21]. The models made by Grases et al. [31] show a shift in the growing importance of SLR-induced flooding from 2050 onwards, and at the same time, longshore sediment transport processes are slowed down while they are enhanced in the cross-shore direction. The intensity of the induced coastal hazards will be a consequence of different parameters related with the beach profile height, beach width, and evolution stage, which will modulate the induced morphodynamic response by storms [45]. However, despite the low DVI obtained, it should be taken into account that the more frequent the erosive processes are, the most fragile the coastal stretch will be. Moreover, if the frequency of storms is high during a certain period, it could be possible that natural recovery processes should have not enough time to be effective, and then, storms should impact on already eroded/affected areas [45].

### *4.5. Coastal Management Assessment*

Coastal wetlands are unique ecosystems whose sustainability depends on the resilience of the geomorphic and ecological environment [46]. According to Cobani [47], sedimentation and/or coastal erosion are the main processes that must be controlled to ensure the physical stability of coastal lagoons. Coastal dunes provide a buffer against coastal hazards such as wind erosion, wave overtopping, and tidal inundation during storm events [6]. Therefore, the high resilience and the low DVI values obtained for the Riumar dune could ensure the continuity of the El Garxal coastal lagoon, as long as the same conditions that have allowed its development can be preserved, that is, that Cape Tortosa will be maintained. Since the 1990s, this area has been considered the most vulnerable of the entire deltaic environment [48].

This wetland is of great ecological value as it is not directly affected by the irrigation surplus of the rice crops of the Ebro Delta and is included in the 25% of natural habitats that are still conserved in the Ebro Delta [49]. It is protected by the aforementioned legal figure of the Ebro Delta, in addition to having a particular category of protection as a Wildlife Refuge (since 1989), due to their importance as nesting areas for seabirds and

shorebirds. Likewise, the entire Ebro Delta front is included in the Spanish Inventory of Places of Geological Interest (LIG) [50], as an international geosite (https://info.igme.es/ielig/LIGInfo.aspx?codigo=CAT320 (accessed on 10 May 2021)), given that it presents multiple morphologies whose dynamics make the dune systems unique within the Spanish coast.

The results of the present study on the factors affecting the vulnerability of the dune complex aim to reveal the current problems for the future sustainability of this important delta, as well as which parameters require further attention. It is an effective tool for the management of this coastal protected area. Vulnerability factor analyses have been carried out in many dune systems around the world since the 1990s, using these indices for coastal zone management [51–53]. However, data replicability and comparisons between different sites is complicated, as there are no clear-cut guidelines about how to acquire/collect the data that will be used to calculate the DVI [54]. Standardized and replicable protocols and a multidisciplinary approach to exploit every surveying/analysis technique to match and compare all the data acquired from different sources would be desirable [55].

## 5. Conclusions

The results obtained show that the Riumar dune field in El Garxal coastal wetland presents low vulnerability, and is able to withstand the most severe conditions as long as it has enough time for recovery. However, if the spatial scale considered is broadened, the vulnerability of the system increases. The El Garxal coastal barrier-lagoon system is in a highly fragile environment such as the Ebro delta, and coastal flooding and erosion events are expected due to the influence of climate change that would increase the vulnerability of this coastal stretch. It is essential to generate protection efforts towards the most vulnerable areas identified on the coast of the Ebro delta.

**Author Contributions:** Conceptualization, I.R.-S.; methodology, I.R.-S. and A.D.-M.; software, I.R.-S. and A.D.-M.; validation, I.R.-S. and A.D.-M. and N.N.; formal analysis, I.R.-S. and A.D.-M.; investigation, I.R.-S. and A.D.-M.; resources, I.R.-S. and A.D.-M.; data curation, I.R.-S.; writing—original draft preparation, I.R.-S. and A.D.-M.; writing—review and editing, I.R.-S. and A.D.-M. and N.N.; visualization, I.R.-S. and A.D.-M. and N.N.; supervision, I.R.-S.; project administration, I.R.-S. and N.N.; funding acquisition, I.R.-S. and N.N. All authors have read and agreed to the published version of the manuscript.

**Funding:** This research was funded by the Universidad Rey Juan Carlos, Spain.

**Data Availability Statement:** Data available from the authors on request.

**Acknowledgments:** This work was carried out under the project "Estudio integral del sistema barrera arenosa-laguna costera desarrollado en la desembocadura del Ebro como ecosistema centinela para observaciones costeras del cambio global (ICOLADE)" funded by the Universidad Rey Juan Carlos, Spain.

**Conflicts of Interest:** The authors declare no conflict of interest.

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
