# Peer review of "Vulnerability Analysis of the Riumar Dune Field in El Garxal Coastal Wetland (Ebro Delta, Spain)"

_jmse, doi:10.3390/jmse9060601_

Round 1

Reviewer 1 Report

Dunes in deltaic environments are really notable objects, investigation of which sheds light on the coastal zone dynamics. The reviewed manuscript presents novel information from the assessment of the dune filed in the Ebro River delta. It is based on a sound project, and the outcomes will be interesting to learn to the international research community. The manuscript is generally well-structured and well-written. Indeed, it needs some additional work, including several extensions, and I indicate below how the authors may achieve this task.

  • Title & abstract: please, indicate briefly (with 1-2 words) vulnerability to what.
  • Keywords: please, avoid words from the title.
  • Introduction: please, move study area to the next, methodological section.
  • 1: wind and wave rose diagrams should be shown below the other drawings to allow their higher resolution and better visibility.
  • If I'm not mistaken subsections 4.1 and 4.2 are named similarly.
  • Discussion: In my opinion, the authors need to state implications of their findings to the regional land management. I also encourage the authors to think about attribution of this dune field to geoheritage and to write what do the findings mean to geoheritage conservation. This literature may help (see also citations there):

https://www.mdpi.com/2076-3298/6/2/18

https://www.mdpi.com/2071-1050/12/17/7109/htm

  • Discussion: I think putting all these findings in the broader frame of the international research will strengthen this paper. Particularly, I encourage to compare the results of this study to those of the similar studies in some other places of the world. I also think that the great idea of 'maintaining itself' should be conceptualized and supported with some other examples from the literature.
  • The writing is clear, but I suggest to avoid too short paragraphs.

Author Response

Dear reviewer,

We greatly appreciate your comments, they are are very helpful and important for us.In the attached file you can find the detailed response point by point.

Reviewer 2 Report

This study presents some insights to implement a built-in vulnerability in the EL Garxal dune field. The proposal presents scientific relevance in the current framework of anthropic and environmental changes. In my opinion, the manuscript can be improved in some aspects, which I detail below:

The authors perform a vulnerability analysis based on 19 variables that indicate moderate susceptibility and resilience. The previous works on which this evaluation is based, instead show around 50 variables. In particular García Mora 2001 amount of 53 variables, and Peña-Alonso 2018 a total of 42. Also, in the work of Ciccarelli 2017 in the Mediterranean area with a total of 51 variables derived. Could the authors justify what adjustments and integrations were introduced with respect to previous works? Why these 19 variables may be enough in the Garxal dune field to estimate the Vulnerability Index?

Fig. 1 Wave Roses are not readable. Check image resolution and sign size

In the study area, it is stated that in recent decades drastic geomorphologic changes occurred, as shown in figure 2 and Rodríguez-Santalla, I.; Triviño-Monozón, J. 2019 references. Could the authors update Figure 2 to include the most recent changes? This will improve the understanding of this work

Storm damage occurred due to Gloria storm on 19 and 23 January of 2020, where a Hs of 7.6 m and H max of 12 m was reached, is described. However, are not quantify any geomorphological changes. On the other hand, the other stormy episode that took place, the Filomena storm, is not described. For example, the significant height and period of the storm. If possible, the authors should quantify the storm damage or include suitable references.

Section 2. Materials and Methods should be better detailed. Variables such as Marine influence (MI), Human pressure (HP), characteristics of the vegetation cover (VC) are poorly described. Each of these item’s descriptions corresponds to a greater number of variables in previous studies and it should also be detailed which is the source and the evaluation criteria in order to calculate the vulnerability index.

 The Aeolian influence (AI) To get the sand supply input analysis of the volume variation has been obtained comparing the MDTs of 2011 and 2016 using the cut and fill tool of ArcGIS. The Net Shoreline Movement between 1998 and 2018 shows a mean value of +160 meters, which is equivalent to 8 meters of advance per year. The fieldwork/data validation has not been presented in the methodology either. Authors should make an effort to improve the description of the methods section.

 Finally, I suggest the authors include a conclusions section to summarize and highlight the scientific advances this manuscript presents.

Author Response

(The authors gave the same response as above.)

Round 2

Reviewer 1 Report

The authors did a good job, and the revised version of their manuscript looks much better. They responded all my questions (although a big differently than I imagined in one case, but this is not a problem at all). It addresses an important object and appealing research question, which is resolved elegantly with a state-of-the-art analysis. The manuscript itself is well-written and well-structured. Undoubtedly, it will be interesting to the broad circle of researchers.

I see one minor technical issue. In References, the item 53 includes TWO sources. Please, select one, and if the both are necessary cite the other one with a different number. Please, control the relevant citations in the text.

Author Response

Dear reviewer:

We are grateful to you for allocating time to review our manuscript and we appreciate your constructive comments and suggestions which has improved the paper.

A new revision of the references has been made, and some errors have been corrected.

Reviewer 2 Report

The authors have reviewed the manuscript,
Can be published in its present form.

Author Response

Dear reviewer:

We are grateful to you for allocating time to review our manuscript and we appreciate your constructive comments and suggestions which has improved the paper.